# Accelerometer measured physical activity and the incidence of cardiovascular disease: Evidence from the UK Biobank cohort study

Rema Ramakrishnan[1,2], Aiden Doherty[3,4,5], Karl Smith-Byrne[6], Kazem Rahimi[1,5,7,8], Derrick Bennett[5,9], Mark Woodward[10,11,12], Rosemary Walmsley[3,4], Terence Dwyer[1,13]*

**1** Nuffield Department of Women's and Reproductive Health, University of Oxford, Oxford, United Kingdom, **2** University of New South Wales, Sydney, Australia, **3** Big Data Institute, Li Ka Shing Centre for Health Information and Discovery, University of Oxford, Oxford, United Kingdom, **4** Nuffield Department of Population Health, University of Oxford, Oxford, United Kingdom, **5** National Institute of Health Research Oxford Biomedical Research Centre, Oxford University Hospitals NHS Foundation Trust, John Radcliffe Hospital, Oxford, United Kingdom, **6** International Agency for Research on Cancer, Genetic Epidemiology Group, Lyon, France, **7** Deep Medicine, Oxford Martin School, University of Oxford, United Kingdom, **8** Oxford University Hospitals NHS Foundation Trust, Oxford, United Kingdom, **9** Clinical Trials Service Unit and Epidemiological Studies Unit (CTSU), Nuffield Department of Population Health, University of Oxford, Oxford, United Kingdom, **10** The George Institute for Global Health, Nuffield Department of Women's and Reproductive Health, University of Oxford, Oxford, United Kingdom, **11** The George Institute for Global Health, University of New South Wales, Sydney, Australia, **12** Department of Epidemiology, Johns Hopkins University, Baltimore, United States of America, **13** Murdoch Children's Research Institute, Melbourne, Australia

* terence.dwyer@wrh.ox.ac.uk

## Abstract

### Background

Higher levels of physical activity (PA) are associated with a lower risk of cardiovascular disease (CVD). However, uncertainty exists on whether the inverse relationship between PA and incidence of CVD is greater at the highest levels of PA. Past studies have mostly relied on self-reported evidence from questionnaire-based PA, which is crude and cannot capture all PA undertaken. We investigated the association between accelerometer-measured moderate, vigorous, and total PA and incident CVD.

### Methods and findings

We obtained accelerometer-measured moderate-intensity and vigorous-intensity physical activities and total volume of PA, over a 7-day period in 2013–2015, for 90,211 participants without prior or concurrent CVD in the UK Biobank cohort. Participants in the lowest category of total PA smoked more, had higher body mass index and C-reactive protein, and were diagnosed with hypertension. PA was associated with 3,617 incident CVD cases during 440,004 person-years of follow-up (median (interquartile range [IQR]): 5.2 (1.2) years) using Cox regression models. We found a linear dose–response relationship for PA, whether measured as moderate-intensity, vigorous-intensity, or as total volume, with risk of incident of CVD. Hazard ratios (HRs) and 95% confidence intervals for increasing quarters

**Data Availability Statement:** The data underlying the results presented in the study are available from the UK Biobank. The data are available to

researchers through a procedure described at http://www.ukbiobank.ac.uk/using-the-resource/.

**Funding:** AD is supported by the National Institute for Health Research (NIHR) Oxford Biomedical Research Centre (BRC), the Alan Turing Institute and the British Heart Foundation (grant number SP/18/4/33803), and Health Data Research UK, an initiative funded by UK Research and Innovation, Department of Health and Social Care (England) and the devolved administrations, and leading medical research charities. RW is supported by a Medical Research Council Industrial Strategy Studentship [grant number MR/S502509/1]. Computation used the Oxford Biomedical Research Computing (BMRC) facility, a joint development between the Wellcome Centre for Human Genetics and the Big Data Institute supported by Health Data Research UK and the NIHR Oxford Biomedical Research Centre. The views expressed are those of the author(s) and not necessarily those of the NIHR. KR is supported by the National Institute of Health Research (NIHR) Oxford Biomedical Research Centre. KR further receives grants from the Oxford Martin School, University of Oxford as well as the PEAK Urban programme from the UKRI's Global Challenge Research Fund Grant Ref: ES/P011055/1 and the British Heart Foundation. The funders had no role in study design, data collection and analysis, decision to publish, or preparation of the manuscript.

**Competing interests:** I have read the journal's policy and the authors of this manuscript have the following competing interests: KR has in the past received personal fees as Associate Editor for PLOS Medicine and as Associate Editor, is in receipt of Personal Fees from BMJ Heart. MW is a consultant to Amen and Kirin. RW is supported by a Medical Research Council Industrial Strategy Studentship (grant number MR/S502509/1). The Medical Research Council had no role in the study design; collection, analysis, and interpretation of data; writing of the paper; and/or decision to submit for publication.

**Abbreviations:** BMI, body mass index; CI, confidence interval; CVD, cardiovascular disease; HbA1c, glycated haemoglobin; HDL, high-density lipoprotein; HR, hazard ratio; ICD, International Classification of Diseases; IHD, ischaemic heart disease; IQR, interquartile range; LDL, low-density lipoprotein; PA, physical activity; STROBE, Strengthening the Reporting of Observational Studies in Epidemiology.

of the PA distribution relative to the lowest fourth were for moderate-intensity PA: 0.71 (0.65, 0.77), 0.59 (0.54, 0.65), and 0.46 (0.41, 0.51); for vigorous-intensity PA: 0.70 (0.64, 0.77), 0.54 (0.49,0.59), and 0.41 (0.37,0.46); and for total volume of PA: 0.73 (0.67, 0.79), 0.63 (0.57, 0.69), and 0.47 (0.43, 0.52). We took account of potential confounders but unmeasured confounding remains a possibility, and while removal of early deaths did not affect the estimated HRs, we cannot completely dismiss the likelihood that reverse causality has contributed to the findings. Another possible limitation of this work is the quantification of PA intensity-levels based on methods validated in relatively small studies.

## Conclusions

In this study, we found no evidence of a threshold for the inverse association between objectively measured moderate, vigorous, and total PA with CVD. Our findings suggest that PA is not only associated with lower risk for of CVD, but the greatest benefit is seen for those who are active at the highest level.

## Author summary

### Why was this study done?

- There is an inverse association between self-reported physical activity (PA) and occurrence of cardiovascular disease (CVD). However, there is uncertainty about the strength of this association as self-reported questionnaires are subject to differential measurement error.

- Accelerometers are small lightweight motion sensors that are typically worn on the wrist. They more reliably measure PA status and thus could improve understanding of associations with CVD.

### What did the researchers do and find?

- We used data from 90,211 UK Biobank participants without prior CVD who agreed to wear an accelerometer to measure their PA status over 7 days.

- Higher levels of PA were associated with lower risk for CVD that was similar across total, moderate- and vigorous-intensity PA.

### What do these findings mean?

- Our findings suggest that individuals who engage in higher levels of PA have lower risk for CVD throughout the range of PA measured.

- The lowest risk for CVD in the UK Biobank cohort is seen at the highest level of PA, whether total, moderate-intensity, or vigorous-intensity.

## Introduction

Higher levels of physical activity (PA) have been found to be inversely associated with the risk of cardiovascular disease (CVD) [1–5]. However, most of this evidence has relied on questionnaires that measure PA relatively imprecisely and with different validity in population subgroups defined by age, sex, and socioeconomic status [6]. Further, questionnaire-based methods are not well suited to capture incidental activity that occurs throughout the day and therefore do not validly measure all movement that occurs in a specified time period [7]. This could, therefore, lead to uncertainty in the estimation of the strength of the association between PA and CVD and in estimating its shape, i.e., are increasing levels of activity associated with additional cardiovascular benefit?

Objective measures, such as wrist-worn accelerometers, are able to incorporate all components of PA—frequency, intensity, and duration—as a continuous score and thus validly capture all PA undertaken [7]. This is likely to be important since total energy expenditure is conceivably the primary pathway through which PA reduces risk of disease [8]. While an increasing number of studies have examined the association of objectively measured PA and mortality, few have assessed its association with incident CVD [9–12]. We investigated the association of moderate-intensity and vigorous-intensity PAs and total volume of PA, measured objectively by accelerometer, with incident CVD in 90,211 UK Biobank participants.

## Methods

The UK Biobank, a large population-based cohort study, was established to enable research into genetic and nongenetic risk factors for diseases of middle and old age through longitudinal follow-up of participants throughout the UK. The UK Biobank recruited over 500,000 people aged between 40 and 69 years in 2006 to 2010 from across the UK. These participants provided blood, urine, and saliva samples for future analysis and detailed information about themselves and agreed to be followed for multiple health-related outcomes [13–15]. Ethical approval was obtained by the UK Biobank from the North West Multicentre Research Ethics Committee, the National Information Governance Board for Health and Social Care in England and Wales, and the Community Health Index Advisory Group in Scotland. All participants provided written informed consent.

### Assessment of physical activity

Data from a subsample of 103,687 participants who wore an Axivity AX3 triaxial accelerometer on their dominant wrist were collected over a 7-day period in 2013 to 2015. It has been found that wrist-worn accelerometers can explain about 44% to 47% of the variation in PA energy expenditure as measured by doubly labelled water [16]. We only included participants whose accelerometer data could be successfully calibrated, meaning that different devices provide comparable data outputs [14]. We excluded participants with >1% clipped values, which occur when the sensor's dynamic range of ±8 g is exceeded before or after calibration [14]. In addition, we excluded participants with implausibly high activity values, defined as average vector magnitude scores of >100 mg. We also excluded participants with insufficient wear-time, defined as the unavailability of at least 72 hours of data or who lacked data for every 1-hour period of the 24-hour cycle (scattered over multiple days) [14]. This resulted in 96,675 participants with accelerometer data for the analysis (see S1 Fig for flowchart of study participants).

We extracted total volume of PA, measured as the average vector magnitude in milli-gravity (mg) units. The metric, average vector magnitude, has demonstrated good face validity in the

UK Biobank (activity: 7.5% lower per decade of age) [14] and has been validated against doubly labelled water which is a gold-standard measure for energy expenditure [16]. We estimated minutes of moderate and vigorous PA per week from percentage of time spent in 100 mg to 400 mg and above 400 mg, respectively [17,18]. The total volume, and each intensity of PA, were categorised into the equal quarters of their distributions in the analytic sample.

## Assessment of cardiovascular disease

Incident CVD was defined as the first hospital admission or death from CVD, defined as ischaemic heart disease (IHD; International Classification of Diseases (ICD)-10 codes: I20-25) or cerebrovascular disease (ICD codes: I60-I69), identified from linkages to the national death index and Hospital Episode Statistics [15]. In secondary analyses, we analysed IHD and cerebrovascular disease separately.

## Statistical analysis

This study did not have a prespecified analysis plan, and no data-driven analyses were included.

We excluded participants who had been diagnosed with CVD from follow-up hospital records, before the end of their accelerometer wear (prevalent cases) resulting in 91,040 participants (S1 Fig). Follow-up time was calculated as person-time in months for each participant from the final date of accelerometer wear to the first occurrence of CVD or the end of study (31 March 2020, 29 February 2016, and 31 October 2016 for participants from England, Wales, and Scotland, respectively). The analytic sample consisted of 90,211 participants who had complete data for PA, age, sex, ethnicity, age completed full time education, Townsend Deprivation Index, smoking, and alcohol consumption.

We computed descriptive statistics—mean (standard deviation)/median (interquartile range (IQR)) for continuous measures and percentage for categorical variables—by categories of total volume of PA. We used multivariable-adjusted Cox proportional hazards regression models to estimate hazard ratios (HRs) for the association between total volume, moderate- and vigorous PAs, and risk of CVD. Cox regression was also used for subgroup and sensitivity analyses. Analyses were adjusted for age (stratified by 5-year age-at-risk intervals to satisfy the proportional hazards assumption within each age-at-risk group), sex, ethnicity (white and nonwhite), age completed full time education (years), area-based social deprivation using the Townsend Deprivation Index [19] (categorised into quarters), smoking (never, former, and current), and alcohol (never, less than 3 times/week, and ≥3 times/week) which were collected at baseline (2006 to 2010). The proportional hazards assumption was assessed using log (-log) survival plots and covariate-by-(log) time interaction terms [20], and none of the analysed variables violated this assumption. We conducted competing risk analysis using cause-specific and Fine and Gray methods [21]. There was no material difference between the estimates from these models and the models that did not account for competing risks. Therefore, we have presented the results from the latter.

We assessed the shape of the relationship between moderate-intensity and vigorous-intensity, total volume of PA, and incident CVD using a restricted cubic spline model. For this purpose, we trimmed observations less than 5% and greater than 95% of the distribution. We specified the knots at the 25th, 50th, and 75th centiles that were used for the categorisation of the variables for total volume, moderate-intensity, and vigorous-intensity of PA.

To address reverse causation, we repeated the Cox regression analyses after removing CVD events that occurred within the first year and then 2 years of follow-up. We conducted sensitivity analysis by adjusting for 4 groups of comorbid conditions that may affect participants'

ability to engage in PA—cancer (ICD codes: C01-C26, C30-C58, C60-C97, and D00-D48), diabetes mellitus (ICD codes: E10-E14), hypertension (ICD code: I10), and chronic lower respiratory disease (ICD codes: J43 and J44.9). We conducted prespecified subgroup analysis by sex. For subgroup analyses by sex, we categorised total volume and each intensity of PA into equal quarters of their distributions that were sex-specific. Furthermore, we examined the change in HR for each level of PA and CVD after sequential adjustment of variables not included in the main Cox regression analyses. These variables were hypertension (ICD code: I10); self-rated health (good/excellent versus fair/poor); body mass index (BMI) (categorised as underweight ($<$18.5 kg/m$^2$), normal weight (18.5 to 24.9 kg/m$^2$), overweight (25.0 to 29.9 kg/m$^2$), and obese ($\geq$30.0 kg/m$^2$)); total cholesterol (mmol/L), high-density lipoprotein (HDL, mmol/L); low-density lipoprotein (LDL, mmol/L); triglycerides (mmol/L); log-transformed C-reactive protein (mg/L); glycated haemoglobin (HbA1c) (mmol/mol) categorised into $<$42 mmol/mol, 42 to 47 mmol/mol, and $\geq$48 mmol/mol; red and processed meat (times/week); fresh fruit (serving/day); and cooked vegetable (serving/day).

Data were analysed using SAS 9.4 (SAS Institute, Cary, North Carolina, United States of America) and STATA 14.0 (StataCorp. 2015, College Station, Texas, USA), and statistical testing was conducted at a 2-tailed alpha level of 0.05.

This study was reported as per the Strengthening the Reporting of Observational Studies in Epidemiology (STROBE) guidelines [22] (S1 STROBE Checklist).

## Results

For the 90,211 participants included in the current study, 3,617 were diagnosed with CVD (3,305 nonfatal and 312 fatal) during 440,004 person-years of follow-up (median (IQR): 5.2 (1.2) years). Out of 6,614 participants without prevalent CVD who were excluded due to poor wear time, 262 (4.0%) were diagnosed with incident CVD during a median follow-up of 62.6 months (compared with 4.0% cases during a median follow-up of 61.9 months for the study participants). Participants in the highest category of total PA engaged in more moderate and vigorous accelerometry-measured PA. Compared with participants in the highest category of total PA, participants in the lowest category also had a higher BMI, smoked more, drank slightly more alcohol, were diagnosed with hypertension, and had higher levels of C-reactive protein and HbA1c (Table 1).

We found a linear dose–response relationship between moderate and vigorous PAs and risk of incident CVD (Figs 1, 2B and 2C). The findings for total PA were very similar (Figs 1 and 2A). Compared with the lowest category of moderate-intensity PA, the HRs and 95% CIs for increasing quarters were 0.71 (0.65, 0.77), 0.59 (0.54, 0.65), and 0.46 (0.41, 0.51) (Fig 1), and the corresponding values for vigorous activity were 0.70 (0.64, 0.77), 0.54 (0.49, 0.59), and 0.41 (0.37, 0.46). We found similar trends in HRs for total PA and incident CVD (Fig 1).

### Subgroup, secondary, and sensitivity analyses

During the study period, 2,220 men and 1,397 women developed CVD. There were minimal differences between the sexes in risk of CVD for moderate and total volume of PAs. However, for vigorous PA, compared with men, a stronger inverse association was observed for women (Fig 3) ($P_{\text{interaction}} < 0.001$) especially for second quarter of vigorous PA compared with the first.

For analyses by type of CVD, there were 2,773 participants who were diagnosed with IHD and 844 with cerebrovascular disease. The HRs by categories of moderate, vigorous, and total PAs did not differ substantially from the overall CVD estimates (Fig 4).

**Table 1. Baseline characteristics of the participants in 2006–2010 by quarters of accelerometer-measured total volume of PA ($N$ = 90,211).**

| | ≤22.7 mg | 22.8–27.3 mg | 27.4–32.7 mg | >32.7 mg | Total |
|---|---|---|---|---|---|
| Follow-up, months, median (IQR) | 61.5 (14.5) | 62.0 (14.0) | 62.0 (14.0) | 62.5 (13.9) | 61.9 (14.1) |
| **CVD event**, $n$ (%) | 1,385 (6.1) | 924 (4.1) | 758 (3.4) | 550 (2.4) | 3,617 (4.0) |
| **Age at accelerometer measurement**—mean (SD) | | | | | |
| | 64.6 (7.4) | 62.7 (7.7) | 61.3 (7.7) | 59.3 (7.6) | 62.0 (7.8) |
| **PA-related factors** (min/week)—mean (SD)[a] | | | | | |
| Moderate activity | 405.6 (124.2) | 628.9 (103.0) | 816.7 (124.5) | 1,122.0 (261.6) | 743.2 (310.6) |
| Vigorous activity | 10.1 (10.1) | 10.1 (10.1) | 20.2 (30.2) | 50.4 (60.5) | 20.2 (30.2) |
| **Sex** (%) | | | | | |
| Female | 11,831 (52.4) | 13,212 (58.6) | 13,600 (60.4) | 13,548 (60.1) | 52,191 (57.9) |
| Male | 10,736 (47.6) | 9,348 (41.4) | 8,932 (39.6) | 9,004 (39.9) | 38,020 (42.1) |
| **Age completed full time education**—mean (SD) | | | | | |
| | 18.5 (2.5) | 18.7 (2.4) | 18.8 (2.4) | 18.8 (2.4) | 18.7 (2.4) |
| **Area deprivation** (%) | | | | | |
| Least deprived (Q1) | 5,378 (23.8) | 5,761 (25.5) | 5,829 (25.9) | 5,622 (24.9) | 22,590 (25.0) |
| Second quarter | 5,582 (24.7) | 5,703 (25.3) | 5,623 (25.0) | 5,645 (25.0) | 22,553 (25.0) |
| Third quarter | 5,633 (25.0) | 5,597 (24.9) | 5,628 (25.0) | 5,647 (25.0) | 22,505 (25.0) |
| Most deprived (Q4) | 5,974 (26.5) | 5,499 (24.4) | 5,452 (24.2) | 5,638 (25.0) | 22,563 (25.0) |
| **Ethnicity** (%) | | | | | |
| White | 22,025 (97.6) | 21,913 (97.1) | 21,812 (96.8) | 21,707 (96.2) | 87,457 (97.0) |
| Nonwhite | 542 (2.4) | 647 (2.9) | 720 (3.2) | 845 (3.8) | 2,754 (3.0) |
| **Smoking status** (%) | | | | | |
| Never | 12,304 (54.5) | 13,053 (57.9) | 13,357 (59.3) | 13,561 (60.1) | 52,275 (58.0) |
| Former | 8,296 (36.8) | 8,007 (35.4) | 7,803 (34.6) | 7,661 (34.0) | 31,767 (35.2) |
| Current | 1,967 (8.7) | 1,500 (6.7) | 1,372 (6.1) | 1,330 (5.9) | 6,169 (6.8) |
| **Alcohol drinking frequency** (%) | | | | | |
| Never | 1,501 (6.6) | 1,164 (5.2) | 1,133 (5.0) | 1,156 (5.1) | 4,954 (5.5) |
| <3× per week | 10,217 (45.3) | 11,174 (49.5) | 11,305 (50.2) | 11,464 (50.8) | 44,160 (48.9) |
| ≥3× per week | 10,849 (48.1) | 10,222 (45.3) | 10,094 (44.8) | 9,932 (44.0) | 41,097 (45.6) |
| **Self-rated health** (%) | | | | | |
| Fair/poor | 5,762 (25.6) | 3,808 (16.9) | 3,117 (13.9) | 2,437 (10.8) | 15,124 (16.8) |
| Good/excellent | 16,743 (74.4) | 18,708 (83.1) | 19,389 (86.1) | 20,089 (89.2) | 74,929 (83.2) |
| **Hypertension** (%) | 548 (2.4) | 424 (1.9) | 352 (1.6) | 251 (1.1) | 1,575 (1.8) |
| **BMI** (kg/m$^2$)—mean (SD) | 28.3 (5.1) | 26.8 (4.4) | 26.1 (4.1) | 25.1 (3.7) | 26.6 (4.5) |
| **BMI category** (kg/m$^2$) (%) | | | | | |
| Underweight (<18.5) | 83 (0.4) | 94 (0.4) | 127 (0.6) | 220 (1.0) | 524 (0.6) |
| Normal (18.5–24.9) | 5,947 (26.5) | 8,302 (36.9) | 9,611 (42.7) | 11,979 (53.2) | 35,839 (39.8) |
| Overweight (25.0–29.9) | 9,628 (42.8) | 9,697 (43.0) | 9,398 (41.8) | 8,147 (36.2) | 36,870 (41.0) |
| Obese (≥30.0) | 6,826 (30.4) | 4,431 (19.7) | 3,361 (14.9) | 2,185 (9.7) | 16,803 (18.7) |
| **Cholesterol** (mmol/L)—mean (SD) | | | | | |
| Total cholesterol | 5.7 (1.2) | 5.8 (1.1) | 5.8 (1.1) | 5.7 (1.0) | 5.8 (1.1) |
| HDL cholesterol | 1.4 (0.4) | 1.5 (0.4) | 1.5 (0.4) | 1.6 (0.4) | 1.5 (0.4) |
| LDL cholesterol | 3.6 (0.9) | 3.6 (0.8) | 3.6 (0.8) | 3.5 (0.8) | 3.6 (0.8) |
| Triglycerides | 1.8 (1.0) | 1.7 (1.0) | 1.6 (0.9) | 1.5 (0.9) | 1.6 (1.0) |
| **C-reactive protein** (mg/L)—mean (SD) | 2.9 (4.5) | 2.3 (3.7) | 2.0 (3.7) | 1.7 (3.3) | 2.2 (3.9) |
| **HbA1c** (mmol/mol) (%) | | | | | |
| <42 | 19,203 (91.2) | 20,009 (95.1) | 20,258 (96.1) | 20,538 (97.3) | 80,008 (94.9) |
| 42–47 | 909 (4.3) | 605 (2.9) | 495 (2.3) | 388 (1.8) | 2,397 (2.8) |
| ≥48 | 955 (4.5) | 433 (2.1) | 335 (1.6) | 193 (0.9) | 1,916 (2.3) |
| **Red and processed meat** (times/week) (%) | | | | | |
| <2 | 6,867 (30.4) | 7,705 (34.2) | 8,108 (36.0) | 8,666 (38.4) | 31,346 (34.8) |
| 2.0–2.9 | 5,754 (25.5) | 5,880 (26.1) | 5,728 (25.4) | 5,591 (24.8) | 22,953 (25.4) |
| 3.0–3.9 | 1,366 (6.1) | 1,279 (5.7) | 1,275 (5.7) | 1,221 (5.4) | 5,111 (5.7) |
| ≥4.0 | 8,576 (38.0) | 7,694 (34.1) | 7,416 (32.9) | 7,071 (31.4) | 30,757 (34.1) |
| **Fresh fruit** (serving/day) (%) | | | | | |

*(Continued)*

**Table 1.** (Continued)

|  | ≤22.7 mg | 22.8–27.3 mg | 27.4–32.7 mg | >32.7 mg | Total |
|---|---|---|---|---|---|
| <2 | 8,503 (37.7) | 7,586 (33.6) | 6,991 (31.0) | 6,524 (28.9) | 29,604 (32.8) |
| 2.0–2.9 | 6,537 (29.0) | 6,698 (29.7) | 6,806 (30.2) | 6,610 (29.3) | 26,651 (29.5) |
| 3.0–3.9 | 4,337 (19.2) | 4,872 (21.6) | 4,871 (21.6) | 5,088 (22.6) | 19,168 (21.3) |
| ≥4.0 | 3,190 (14.1) | 3,404 (15.1) | 3,864 (17.2) | 4,330 (19.2) | 14,788 (16.4) |
| **Cooked vegetable** (serving/day) (%) |  |  |  |  |  |
| <2 | 18,100 (80.2) | 18,011 (79.8) | 18,104 (80.3) | 17,831 (79.1) | 72,046 (79.9) |
| 2.0–2.9 | 3,514 (15.6) | 3,587 (15.9) | 3,431 (15.2) | 3,640 (16.1) | 14,172 (15.7) |
| 3.0–3.9 | 618 (2.7) | 615 (2.7) | 627 (2.8) | 646 (2.9) | 2,506 (2.8) |
| ≥4.0 | 335 (1.5) | 347 (1.5) | 370 (1.6) | 435 (1.9) | 1,487 (1.7) |

[a]For vigorous PA, the values are median and IQR.

Column percentages displayed unless specified otherwise. Some percentages do not add to 100.0 due to rounding.

BMI, body mass index; CVD, cardiovascular disease; HbA1c, glycated haemoglobin; HDL, high-density lipoprotein; IQR, interquartile range; LDL, low-density lipoprotein; PA, physical activity; SD, standard deviation.

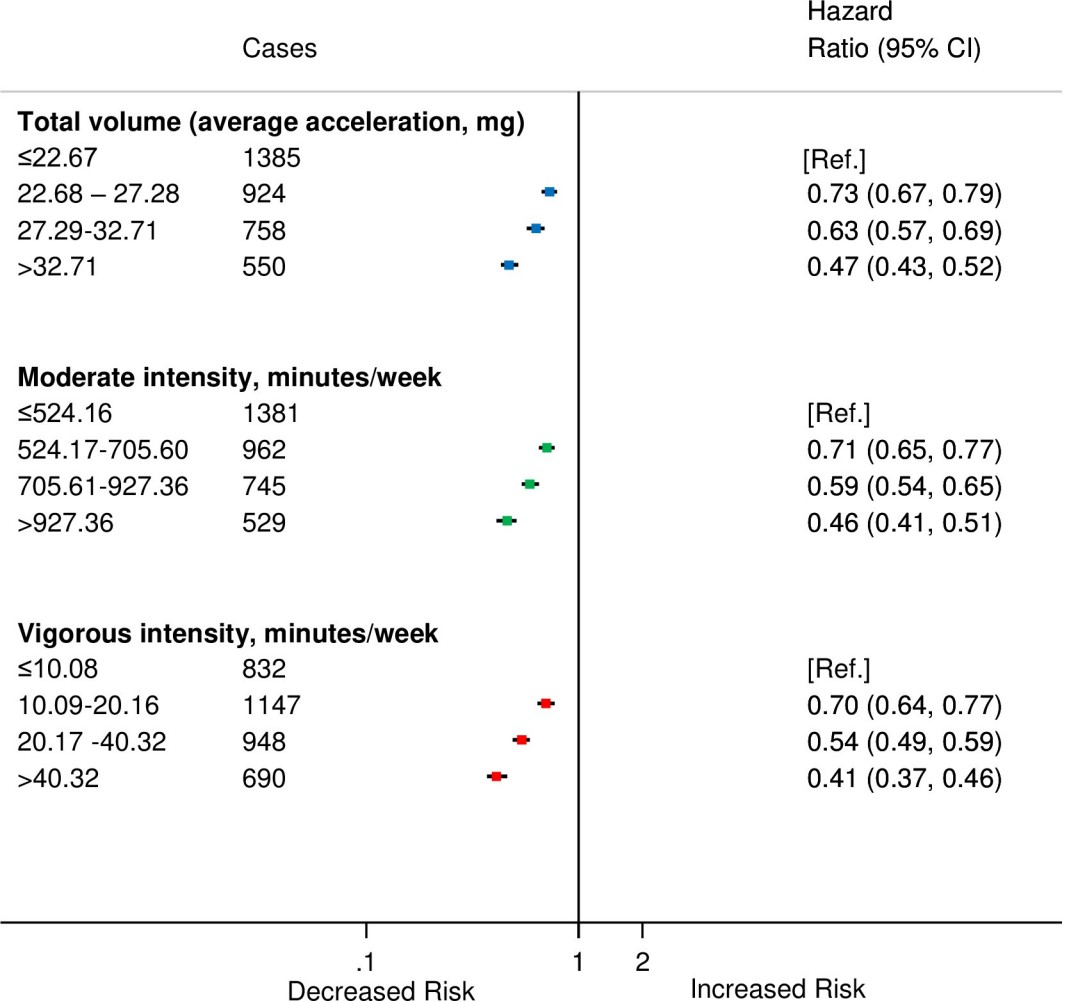

| | Cases | | Hazard Ratio (95% CI) |
|---|---|---|---|
| **Total volume (average acceleration, mg)** | | | |
| ≤22.67 | 1385 | | [Ref.] |
| 22.68 – 27.28 | 924 | | 0.73 (0.67, 0.79) |
| 27.29-32.71 | 758 | | 0.63 (0.57, 0.69) |
| >32.71 | 550 | | 0.47 (0.43, 0.52) |
| **Moderate intensity, minutes/week** | | | |
| ≤524.16 | 1381 | | [Ref.] |
| 524.17-705.60 | 962 | | 0.71 (0.65, 0.77) |
| 705.61-927.36 | 745 | | 0.59 (0.54, 0.65) |
| >927.36 | 529 | | 0.46 (0.41, 0.51) |
| **Vigorous intensity, minutes/week** | | | |
| ≤10.08 | 832 | | [Ref.] |
| 10.09-20.16 | 1147 | | 0.70 (0.64, 0.77) |
| 20.17 -40.32 | 948 | | 0.54 (0.49, 0.59) |
| >40.32 | 690 | | 0.41 (0.37, 0.46) |

Decreased Risk        Increased Risk

**Fig 1. HRs[a] for incident CVD by quarters of average accelerometer-measured total volume, moderate-intensity and vigorous-intensity physical activities in 90,211 UK Biobank participants.** [a]Adjusted for age (stratified by 5-year age-at-risk intervals), sex, ethnicity, education, Townsend Deprivation Index, smoking, and alcohol consumption. CI, confidence interval; CVD, cardiovascular disease; HR, hazard ratio.

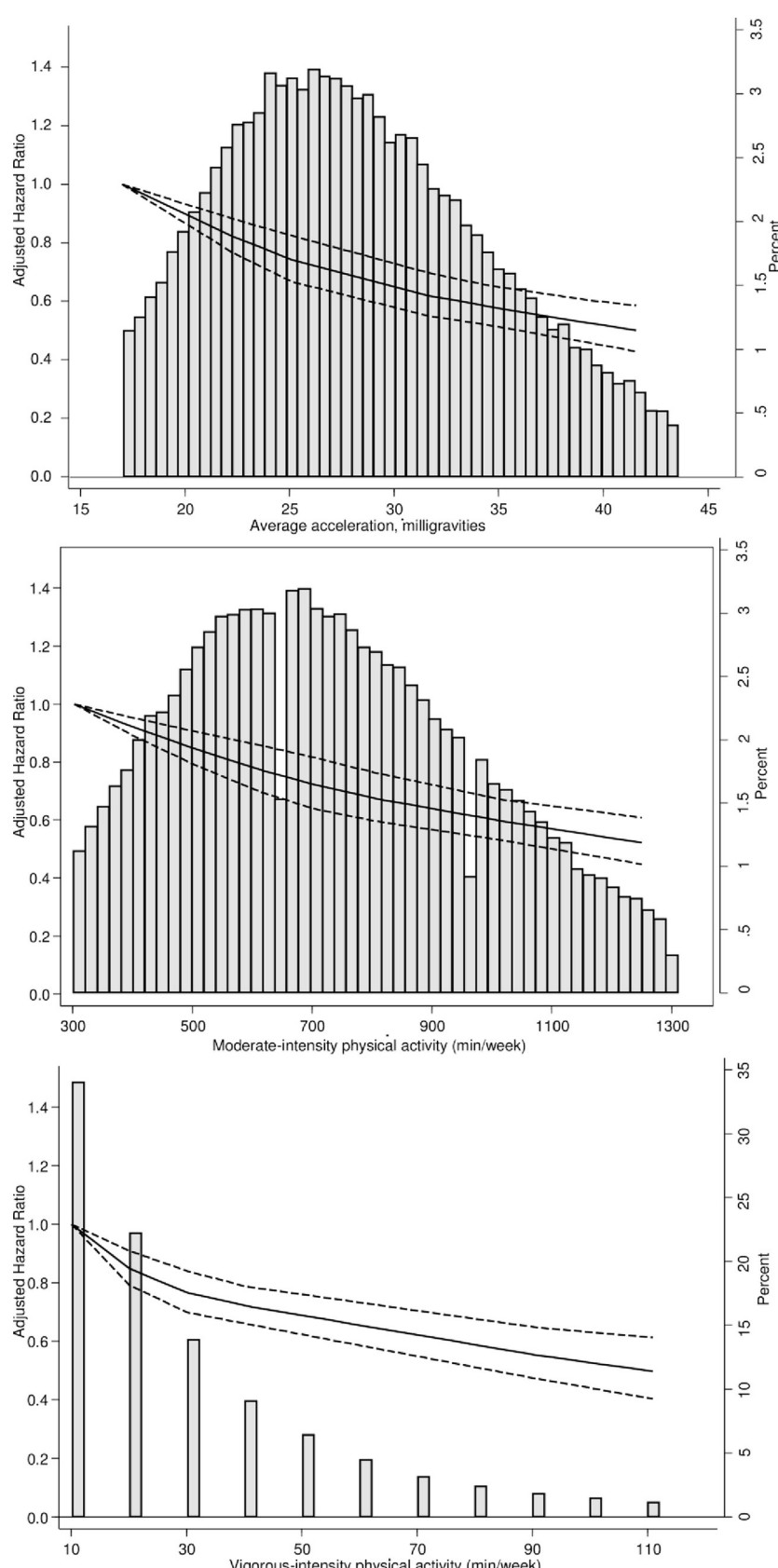

**Fig 2.** Dose–response association (HRs and associated 95% confidence interval band) between accelerometer-measured (A) total volume of PA, (B) moderate-intensity PA, and (C) vigorous-intensity PA and incident CVD using restricted cubic splines with knots at 25th, 50th, and 75th centiles of the distribution of PA (reference category = 17 milligravities (mg) for total volume of PA; 302. 4 minutes/week for moderate intensity PA; 10.08 week for vigorous intensity PA). Also shown are histograms of PA for total volume of PA in milligravities and for moderate-intensity and vigorous-intensity PA in minutes/week. CVD, cardiovascular disease; HR, hazard ratio; PA, physical activity.

During the first year of follow-up, 607 participants were diagnosed with CVD. After excluding participants during the first year of follow-up, there were minimal changes to the HRs (S1 Table). Furthermore, 1,244 participants were diagnosed with CVD within first 2 years of follow-up. After removal of participants within the first 2 years of follow-up, the strong

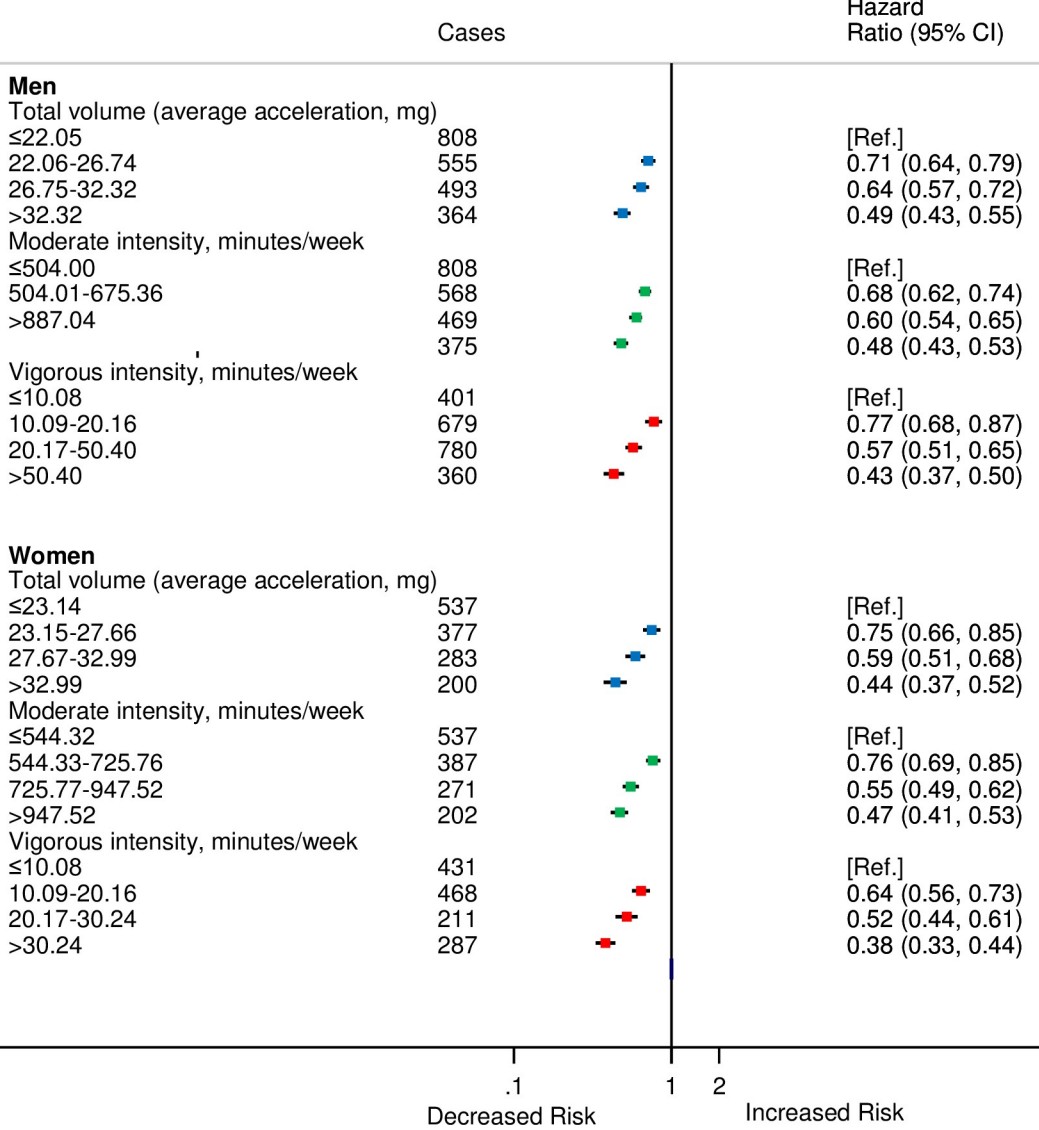

| | Cases | | Hazard Ratio (95% CI) |
|---|---|---|---|
| **Men** | | | |
| Total volume (average acceleration, mg) | | | |
| ≤22.05 | 808 | | [Ref.] |
| 22.06-26.74 | 555 | | 0.71 (0.64, 0.79) |
| 26.75-32.32 | 493 | | 0.64 (0.57, 0.72) |
| >32.32 | 364 | | 0.49 (0.43, 0.55) |
| Moderate intensity, minutes/week | | | |
| ≤504.00 | 808 | | [Ref.] |
| 504.01-675.36 | 568 | | 0.68 (0.62, 0.74) |
| >887.04 | 469 | | 0.60 (0.54, 0.65) |
| | 375 | | 0.48 (0.43, 0.53) |
| Vigorous intensity, minutes/week | | | |
| ≤10.08 | 401 | | [Ref.] |
| 10.09-20.16 | 679 | | 0.77 (0.68, 0.87) |
| 20.17-50.40 | 780 | | 0.57 (0.51, 0.65) |
| >50.40 | 360 | | 0.43 (0.37, 0.50) |
| | | | |
| **Women** | | | |
| Total volume (average acceleration, mg) | | | |
| ≤23.14 | 537 | | [Ref.] |
| 23.15-27.66 | 377 | | 0.75 (0.66, 0.85) |
| 27.67-32.99 | 283 | | 0.59 (0.51, 0.68) |
| >32.99 | 200 | | 0.44 (0.37, 0.52) |
| Moderate intensity, minutes/week | | | |
| ≤544.32 | 537 | | [Ref.] |
| 544.33-725.76 | 387 | | 0.76 (0.69, 0.85) |
| 725.77-947.52 | 271 | | 0.55 (0.49, 0.62) |
| >947.52 | 202 | | 0.47 (0.41, 0.53) |
| Vigorous intensity, minutes/week | | | |
| ≤10.08 | 431 | | [Ref.] |
| 10.09-20.16 | 468 | | 0.64 (0.56, 0.73) |
| 20.17-30.24 | 211 | | 0.52 (0.44, 0.61) |
| >30.24 | 287 | | 0.38 (0.33, 0.44) |

.1 1 2

Decreased Risk Increased Risk

**Fig 3. HRs[a] for incident CVD by quarters of accelerometer-measured total volume, moderate, and vigorous physical activities stratified by sex in 90,211 UK Biobank participants.** [a]Adjusted for age (stratified by 5-year age-at-risk intervals), ethnicity, education, Townsend Deprivation Index, smoking, and alcohol consumption.

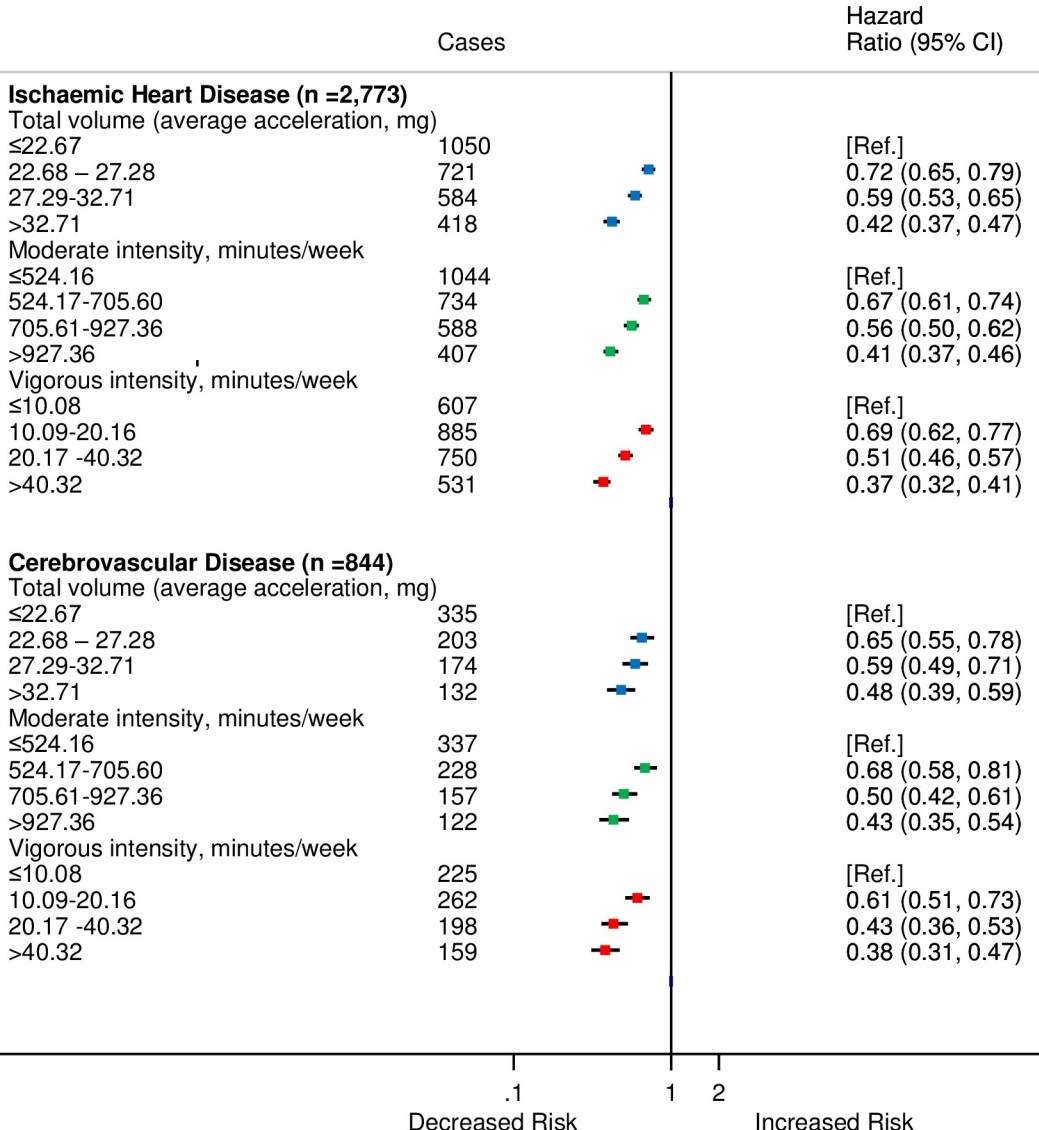

**Fig 4. HRs[a] for incident CVD by quarters of average accelerometer-measured total volume, moderate, and vigorous physical activities for IHD and cerebrovascular disease in 90,211 UK Biobank participants.** [a]Adjusted for age (stratified by 5-year age-at-risk intervals), sex, ethnicity, education, Townsend Deprivation Index, smoking, and alcohol consumption. CI, confidence interval; CVD, cardiovascular disease; HR, hazard ratio; IHD, ischaemic heart disease.

association of total volume of PA with CVD incidence persisted (S1 Table) across both moderate- intensity and vigorous-intensity and total volume of PAs.

The analyses that excluded participants with comorbid conditions such as cancer, diabetes mellitus, hypertension, or chronic lower respiratory disease did not substantially alter the findings (S2 Fig). Furthermore, sensitivity analyses conducted to examine the inclusion of additional covariates (hypertension, self-rated health, BMI, total cholesterol, HDL, LDL, triglycerides, C-reactive protein, HbA1c, red and processed meat, fresh fruit, and cooked vegetables) to the multivariable Cox models utilised in the main analyses attenuated the HRs by just 6.6% to 18.0% (S2–S4 Tables).

## Discussion

In this large population-based cohort study in adults, we found a linear inverse dose–response association between accelerometer-measured moderate and vigorous PA, as well as total PA and incident CVD, with no threshold of effect at low or high levels. We found broadly similar associations within each subcomponent of CVD, and the associations were comparable for men and women, with the exception of vigorous activity where there was a 23% lower risk for incident CVD among men compared with 36% lower risk among women for the second quarter compared to the first.

The inverse association that we found for accelerometer-measured PA and incident CVD is much stronger than that reported from questionnaire-based studies [1,2,23,24]. Notably, we saw no evidence of a higher risk of CVD, particularly stroke, in those engaging in high levels of PA, including vigorous, in contrast to an observation of greater risk in such individuals in 1 large British cohort [25]. In our study, there were minimal differences in the HRs for total, moderate, and vigorous PAs. In contrast to these findings, the protective effect reported by questionnaire-based studies is about 26% for total PA, 20% to 25% for moderate volume/intensity of PA, and 30% to 35% for high amounts/intensity of PA [1,2]. However, the results of the current study concur with the findings from a harmonised meta-analysis with mortality as the outcome that found the protective effect of accelerometer-measured PA to be much greater than that reported in the literature for questionnaire-based studies [10].

We found a linear dose–response association between accelerometer-measured PA and incident CVD with no threshold effect. This is in contrast to the curvilinear dose–response associations observed in studies based on leisure time PA measured via questionnaires [26,27], but similar to those observed in accelerometer-based studies [12,28]. It is noteworthy that Shiroma and Lee [2] found the curvilinear dose–response relationship between self-reported PA and CVD to be present among men whereas in women the shape of the curve appears to be more linear. In contrast, our study found no threshold effect in either men or women.

In our study we found that, using objective measurement, the magnitude of association for total volume of PA was similar to that for moderate and vigorous PAs, which is important for clinical and public health practice. One explanation for this finding is that the participants in the highest level of total volume of PA engaged in higher amounts of moderate-intensity and vigorous-intensity PAs. In support of the contention that the total PA might have the central role in lower risk of CVD, a recent individual-level meta-analysis of 8 studies of accelerometer-assessed PA found that higher levels of total volume of PA irrespective of intensity had an inverse association with mortality from all causes [10]. Future research may need to concentrate on the components of total volume of PA using validated measures of activity intensity [29] in a large sample to unravel the contribution of each.

There is insufficient evidence in the literature to demonstrate that PA confers differential benefits for men and women for CVD. The similarity of findings between men and women for total volume of PA is in accordance with a study from the China Kadoorie Biobank study that used questionnaire data [23]. However, Sattelmair and colleagues found the protective effect to be greater among women than men (36% versus 21%) [1]. Similarly, a review by Shiroma and Lee [2] reported a median risk reduction of 40% and 30%, respectively, for coronary heart disease/CVD among women compared with men, when comparing highest versus lowest total PA. This could conceivably be explained by a different balance of the components of total PA across the different studies. In other words, if moderate and vigorous PA did have different effects on risk, then differences in observed associations of total PA with risk across studies could be explained by a disparity in the contribution of vigorous versus moderate PA to total PA across study samples.

## Strengths and limitations

The major strengths of this study are its large sample size and prospectively collected data on CVD events. In addition, PA was measured objectively by an accelerometer which can capture leisure and non-leisure PAs across multiple domains [28] and minimises the potential for recall bias associated with PA assessed with questionnaires. The UK Biobank cohort consists of mostly white participants living in less socioeconomically deprived areas. Even though the characteristics of the participants may not be generalisable to other populations, it can be used to provide valid estimates of exposure–disease relationships due to its large sample size and multitude of exposures [30,31].

There is currently a lack of robust evidence on whether a 7-day measurement is representative of habitual PA. A previous validation study showed a 7-day measure is strongly associated with PA over a period of up to 3.7 years [9]. In contrast, a recent study found that moderate-to-vigorous PA measured by hip-worn accelerometers is moderately stable over time, but there is considerable within-subject variability [32]. It is therefore possible that if the measurement error is random, our findings are attenuated and have underestimated the true association between PA and CVD.

The mean levels of moderate PA in this study, 743.2 min/week, are much higher than the recommended 150 minutes/week of moderate-to-vigorous activity [33]. However, it is important to note that current guidelines are based on self-reported data on 'sustained' participation in moderate-to-vigorous intensity PA undertaken in bouts of 10 minutes or more. It is suggested that approximately 750 to 1,000 minutes/week of moderate PA is to be expected when using devices that can capture all forms of incidental activity and removing the arbitrary bout criteria [34,35]. In addition, the accurate quantification of levels of moderate PA from accelerometers remains an active challenge, as we have had to rely on methods validated in relatively small validation studies [17,36]. There is a future need for large-scale validation studies in both healthy and diseased individuals [37]. In light of these considerations, we have pragmatically measured mean participant levels of moderate PA in this study.

Given that this is observational data, we also cannot rule out concerns around reverse causality [38] where incipient CVD, not yet detected clinically, might lead to reduced PA because it makes it more difficult for an individual to undertake PA. To address this possibility we removed the first 1 and 2 years following measurement by accelerometry in sensitivity analyses, and the inverse associations for all 3 types of PA persisted. Our adjustment for potentially important confounders (age, sex, ethnicity, education, smoking, alcohol consumption, and deprivation) only had a modest impact on the associations, although we recognise that residual confounding cannot be ruled out entirely.

## Conclusions and implications

In this large population-based cohort, higher levels of moderate- intensity and vigorous intensity PAs as well as total volume were inversely associated with risk of incident CVD with no evidence for a threshold effect. The finding of no threshold effect aligns with the recommendations of the UK Chief Medical Officer's report on PA that "some physical activity is good but more is better" [33].

## Supporting information

**S1 STROBE Checklist.**
(DOCX)

**S1 Table. Adjusted HRs for incident CVD by quarters of average accelerometer-measured total volume (mg), moderate, and vigorous PA after removal of incident CVD occurring within 1 and 2 years of follow-up.** CVD, cardiovascular disease; HR, hazard ratio; PA, physical activity.
(PDF)

**S2 Table. HRs for the association between quarters of total volume of PA (mg) and incident CVD with sequential adjustment for potential confounders and mediators.** CVD, cardiovascular disease; HR, hazard ratio; PA, physical activity.
(PDF)

**S3 Table. HRs for the association between quarters of moderate PA (minutes/week) and incident CVD with sequential adjustment for potential confounders and mediators.** CVD, cardiovascular disease; HR, hazard ratio; PA, physical activity.
(PDF)

**S4 Table. HRs for the association between quarters of vigorous PA (minutes/week) and incident CVD with sequential adjustment for potential confounders and mediators.** CVD, cardiovascular disease; HR, hazard ratio; PA, physical activity.
(PDF)

**S1 Fig. Flowchart of study participants.** Note: The individual numbers for participant exclusion due to low quality accelerometer data do not add to 7,012 because some participants were excluded due to multiple reasons.
(PDF)

**S2 Fig. HRs[a] for incident CVD by quarters of average accelerometer measured total volume, moderate, and vigorous physical activities after excluding participants who had comorbid diseases[b] at baseline.** [a]Adjusted for age (stratified by 5-year age-at-risk intervals), sex, ethnicity, education, Townsend Deprivation Index, smoking, and alcohol consumption. [b]Cancer (ICD codes: C01-C26, C30-C58, C60-C97, and D00-D48), diabetes mellitus (ICD codes: E10-E14), hypertension (ICD codes: I10), and chronic lower respiratory disease (ICD codes: J43 and J44.9). CVD, cardiovascular disease; HR, hazard ratio; ICD, International Classification of Diseases.
(PDF)

## Acknowledgments

This research has been conducted using the UK Biobank Resource under application numbers 15856 and 59070.

## Author Contributions

**Conceptualization:** Terence Dwyer.

**Data curation:** Rema Ramakrishnan.

**Formal analysis:** Rema Ramakrishnan.

**Investigation:** Rema Ramakrishnan, Aiden Doherty, Karl Smith-Byrne, Kazem Rahimi, Derrick Bennett, Mark Woodward, Rosemary Walmsley, Terence Dwyer.

**Methodology:** Rema Ramakrishnan, Aiden Doherty, Karl Smith-Byrne, Kazem Rahimi, Derrick Bennett, Mark Woodward, Terence Dwyer.

**Project administration:** Terence Dwyer.

**Resources:** Aiden Doherty, Terence Dwyer.

**Software:** Rema Ramakrishnan.

**Supervision:** Terence Dwyer.

**Validation:** Rema Ramakrishnan, Aiden Doherty, Karl Smith-Byrne, Kazem Rahimi, Derrick Bennett, Mark Woodward, Rosemary Walmsley, Terence Dwyer.

**Writing – original draft:** Rema Ramakrishnan, Terence Dwyer.

**Writing – review & editing:** Rema Ramakrishnan, Aiden Doherty, Karl Smith-Byrne, Kazem Rahimi, Derrick Bennett, Mark Woodward, Rosemary Walmsley, Terence Dwyer.

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
