## [Editor Report · Decision Letter 0]

11 May 2020

Dear Dr Ramakrishnan, 

Thank you for submitting your manuscript entitled "Accelerometer measured physical activity and the incidence of cardiovascular disease in 90,911 the UK Biobank participants" for consideration by PLOS Medicine.

Your manuscript has now been evaluated by the PLOS Medicine editorial staff [as well as by an academic editor with relevant expertise] and I am writing to let you know that we would like to send your submission out for external peer review.

Kind regards,

Adya Misra, PhD,

Senior Editor

PLOS Medicine

---

## [Decision Letter · Decision Letter 1]

10 Jul 2020

Dear Dr. Dwyer,

Thank you very much for submitting your manuscript "Accelerometer measured physical activity and the incidence of cardiovascular disease in 90,911 the UK Biobank participants" (PMEDICINE-D-20-01924R1) for consideration at PLOS Medicine. 

Your paper was evaluated by a senior editor and discussed among all the editors here, as well as being evaluated by three independent reviewers, including a statistical reviewer. The reviews are appended at the bottom of this email and any accompanying reviewer attachments can be seen via the link below:

[LINK]

In light of these reviews, I am afraid that we will not be able to accept the manuscript for publication in the journal in its current form, but we would like to consider a revised version that addresses the reviewers' and editors' comments. Obviously we cannot make any decision about publication until we have seen the revised manuscript and your response, and we plan to seek re-review by one or more of the reviewers. 

We expect to receive your revised manuscript by Jul 31 2020 11:59PM. Please email us (plosmedicine@plos.org) if you have any questions or concerns.

We look forward to receiving your revised manuscript. 

Sincerely,

Emma Veitch, PhD

PLOS Medicine

On behalf of Clare Stone, PhD, Acting Chief Editor,

PLOS Medicine

plosmedicine.org

*At this stage, we ask that you include a short, non-technical Author Summary of your research to make findings accessible to a wide audience that includes both scientists and non-scientists. The Author Summary should immediately follow the Abstract in your revised manuscript. This text is subject to editorial change and should be distinct from the scientific abstract. Please see our author guidelines for more information: https://journals.plos.org/plosmedicine/s/revising-your-manuscript#loc-author-summary

* In the last sentence of the Abstract Methods and Findings section, please briefly note any key limitation(s) of the study's methodology.

*Please clarify in the Methods section if the analytical approach followed here corresponded to one laid out in a prospective protocol or analysis plan? Please state this (either way) early in the Methods section.

*We noted the authors used the STROBE guideline to help reporting of this study, which is good - if possible, please also state this in the Methods section and cite the corresponding paper/citation (listed here - https://www.equator-network.org/reporting-guidelines/strobe/), and add a mention of the STROBE checklist provided as a supporting information file. 

Comments from the reviewers:

Reviewer #1: The authors present a well-written manuscript examining the association of accelerometer-assessed physical activity and incident cardiovascular disease. Comments below.

Model analysis - age adjustment. Age is noted as stratified within the analysis but not presented by 5-year age groups. Does this mean they were calculated independently for each strata then weighted for an overall measure? Also is there an interaction by age strata compared to an inclusion of the age groupings as a standard covariate within the model?

Fig 2. Distribution component. it is difficult to understand these other than the relative distribution based on the minutes (horizontal axis). it is unclear why A and B are solid areas under the curve, while C is a line. Is this a histogram / cumulative density function? A secondary axis of percent of population may be informative. TThe presentation of the log HR is a bit atypical. is there a reason for not simply using HR, especially as Figure 1 is in this scale. If the authors prefer log HR, please indicate on the curve where the median of each quartile is so we can compare between the two figures.

Fig 3. Shouldnt these quartiles be sex-specific? Did the authors consider using sex-specific quartiles in main analysis (fig 2)?

Suppl Tables 2/3/4. If this is backward/forward selection of variables using a Type 1 p-value, was there an a priori reasoning for the order of the covariates? 

Disc - Dose-response. the authors compare the observed linear association to the Arem LTPA paper which observes a curvilinear association with a threshold effect at 22.5+ met/hr. So both papers show a linear response for the lower 75%. the linear vs threshold is only observed in the highest 25%. Future research may be able to better explore these differences. I am unaware of papers that have looked at the volume reported among the highest active, but the threshold could be due to differential reporting in this group compared to the lower groups. 

Reviewer #2: This is an interesting and useful study on the association between accelerometer measured physical activity and the incidence of cardiovascular disease using the UK Biobank data. The study design, datasets, statistical methods and analyses, and presentation (tables and figures) and interpretation of results are mostly adequate. However, there are still a few issues needing attention.

1) Competing risk. In the survival analyses, incident CVD was used a primary endpoint rather than all cause mortality. Although death from CVD was included, death from other causes became competing risk in the survival analyses. This also applies the secondary analyses for IHD and cerebrovascular disease. Can authors please show the number of deaths from non-CVD causes during the study follow up and perform competing risk analyses as appropriate?

2) Using the 7-day accelerometry measurement of activity status to represent habitual PA is a bit too simplistic. There is not much evidence apart from only one reference. The assumption here is that this 7-day activity represent a person's PA style over the years is subject to scrutiny. This is definitely a limitation and shouldn't be brushed away. Need a bit more comprehensive and critical discussion on this and its potential impact on the results.

3) Throughout the paper, in a few figure legends, it says "Stratified by age-at-risk and adjusted for ethnicity, education, Townsend Deprivation Index, smoking, and alcohol consumption", which seems a bit awkward. Should it simply be "Adjusted for age, ethnicity, education,..."? 

4) Also, in the sensitivity analyses, it is a bit uncomfortable to remove 10,720 participants with either cancer, diabetes mellitus, hypertension, or chronic lower respiratory. Why not just adjusted these 4 conditions in the analyses?

Reviewer #3: This is a well conducted epidemiologic study examined the associations of physical activity measured via accelerometry with incident cardiovascular disease. This study provides important contributions to the literature with its longitudinal design, accelerometer measured physical activity and large sample. 

Other major strengths of this study include that the authors examined physical activity based on various intensities including total PA, moderate and vigorous PA, finding that all intensity had similar linear inverse associations with incident CVD. The authors also included analyses that account for many covariates covering socioeconomic status, other lifestyle behaviors, and co-morbidities. Additionally, the account for reverse causality in sensitivity analyses excluding individuals with CVD events in the first year and 2 years of follow up.

When individuals within the first 2 years of follow-up are excluded the event are substantially smaller. Which demonstrates a major limitation in this study of a relatively short follow up time. There appears to be on average only 2-3 years of follow up. I would suggest continue to emphasize this important limitation throughout the paper.

General Questions/Suggestions:

The additional analyses for reverse causality are helpful (supplement table 1). The number of cases is drastically reduced, so it can be difficult to draw many conclusions, however, it is interesting that the associations for quartile 4, the most active group, is attenuated in model 4. Do the authors believe this is all from reduced sample size and lower events? Could there be a potential that the curvilinear relationship (such as what is discussed by the authors with self-reported PA as the exposure) could exist with longer term follow-up? So in other words, is there potential that with longer follow-up we may see that leveling off/threshold at higher levels? Could only the more immediate risk of developing CVD (e.g. first 2 years of follow-up) to be linear with no leveling off of benefit? It would be interested to hear the authors' perspectives on this.

Another area that would be useful for the authors to address is the quantification of moderate intensity activity and how high these estimates are for this sample. The authors try to address their methodology of how they quantified moderate and vigorous intensity and why they are higher, however, I think the justification could use stronger references. There are more specific comments below on this too. Because the authors use quartiles of total volume of PA, this concern would not influence the results and I believe what the authors did in using quartiles is good way to quantify groups for this study. Additionally, I think it is interesting and relevant to look at both total volume and moderate to vigorous intensity physical activity seperately. However, I would suggest for the authors to be cautious in language about the certainty of how much moderate intensity PA these participants are achieving, unless these values can be confirmed with further references or justifications. There are other cohorts that have objective measures of total MVPA (that includes MVPA accumulated at any bout length) that have much lower levels. The extreme differences appear difficult to justify that this all could be based on differences in device or the cut points used by other cohort studies. For example, supplement 1 of the recent meta analysis by Ekelund et al BMJ 2019 (https://www.bmj.com/content/bmj/suppl/2019/08/19/bmj.l4570.DC1/ekeu048737.ww1.pdf) provide a nice summary of several cohorts and their total MVPA min/day which ranged from 15 to about 37 min/day on average.

Introduction:

For the author's statement that total EE is conceivably the primary pathway through PA reduces risk of disease. Can they provide a better citation than #7 (Bassett et al.) This Bassett et al paper is great for the author's sentence before this on the measurement benefits of examining total PA, however this citation does not adequately justify the link between total EE/volume with disease risk. I would suggest justifying this statement with a stronger citation(s).

Methods:

Page 5: "We only included participants whose accelerometer data could

be successfully calibrated, meaning that data recorded across participants and devices was

comparable". It is unclear what this means. Does this indicate exclusion of individuals with extreme values and the high and low ends of their accelerometer data? And if so, can you further explain how you defined an outlier?

Assessment of physical activity:

I am surprised at how high the duration estimates of moderate intensity activity are for this sample. To demonstrate the validity of using the cut points of 100-400mg - are there other references that can help expand on this? I am not certain the citations provided are adequate. The #16 reference is a paper that includes validation of the ActiGraph and the GENEactiv accelerometers, not Axivity. The #17 reference is a paper that performed a validation of the Axivity among 20 healthy young adults (which is not the best representation of the current sample using in the UK biobank study) of 2 hours of standard behaviors that would not be very comprehensive for activities of daily living. Based on the evidence provided with these citations here, I have reservations on how well these cut points estimate time spent in moderate intensity PA.

Statistical analysis:

Statistical analysis looks good. On page 7 I would suggest adding the following wording to the first sentence for clarity as: "To address reverse causation we repeated the Cox regression analyses after removing [CVD events that occurred in] the first year and then two years of follow-up"

Did the authors do any sensitivity analysis to check for differences between those excluded due to noncompliant accelerometer wear vs. those included? For example, were there higher rates of CVD in the compliant vs. non-compliant accelerometer groups? Any implications for those excluded due to non-compliance in this study?

Results:

Page 8. Also able to estimate this rather quickly based on person years, for transparency and clarity, can the authors add the mean years±s.d. here and/or in the abstract?

Page 8 table 1: the time spent in vigorous PA seems to match up much more closely with accelerometer-measured sum of both moderate AND vigorous PA from other cohort studies. Again, this lead back to my methods concern of how moderate PA was quantified as mentioned above.

Supplement Table 1:

Can the authors please add rows for the number of incident CVD and non-CVD in each quartile that would correspond to models 3 and 4? Knowing the events and total n's in each group would be useful.

Below the table, it reads 'Model 4: Model 3 after removal of first two years…" should this actually be 'Model 4: Model 2 after removal of first two years…"?

Discussion:

Page 13, mid page - the "amounts/intensity" wording is slightly confusing here - can the authors simply use intensity? or does the amount indicate something separately like dose of PA?

Page 14: "using validated measures of activity type" What does type mean here? Do the authors mean intensity?

Page 14: "This could conceivably be explained by a different balance of the components of total PA across the different studies." Could the authors expand on this further for clarity?

Page 15: The authors make good points about the differences of objective measures vs. self report. However, I would suggest removing or taming the last sentence in paragraph 1. "Our study is therefore likely to have more closely measured the activity that participants actually do, rather than what they can perceive what they do". The justifications/citations that the authors currently provide do not appear to adequately justify their estimates of moderate and vigorous activity for their study sample. Reference #27 that the authors provide here is focused on a single study of Actiheart and Bodymedia Armband. The Armband is no longer available and the Actiheart is not as widely used as other accelerometers for large cohort studies. It would be useful to have a few other citations to back up this point. Can the authors further justify/explain how moderate intensity was determined in the validation study? Please see my previous comments from the methods section.

Can anything be said about the generalizability of the UK biobank as a strength or limitation? For instance, is it representative of SES, race/ethnic groups, and health status of the general population? The large sample size certainly helps with this.

It is also worth directly addressing that this study showed that those who do very high volumes of activity are not at an increased risk, and is a good contribution to the literature on this topic given the strength of the accelerometer measures to get objective total volume of activity.

I hope the authors find my feedback useful.

Amanda Paluch

[LINK]

---

## [Decision Letter · Decision Letter 2]

10 Nov 2020

Dear Dr. Dwyer,

Thank you very much for re-submitting your manuscript "Accelerometer measured physical activity and the incidence of cardiovascular disease in 90,211 the UK Biobank participants" (PMEDICINE-D-20-01924R2) for review by PLOS Medicine.

I have discussed the paper with my colleagues and the academic editor and it was also seen again by xxx reviewers. I am pleased to say that provided the remaining editorial and production issues are dealt with we are planning to accept the paper for publication in the journal.

[LINK]

We look forward to receiving the revised manuscript by Nov 17 2020 11:59PM. 

Sincerely,

Adya Misra, PhD

Senior Editor 

PLOS Medicine

plosmedicine.org

Requests from Editors:

Please revise your title according to PLOS Medicine's style. Please place the study design ("A randomized controlled trial," "A retrospective study," "A modelling study," etc.) in the subtitle (ie, after a colon). We suggest “Accelerometer measured physical activity and the incidence of cardiovascular disease in the UK: A Cohort Study”

Abstract- please provide brief participant demographics and please add study dates 

Please provide p-values where needed

Please explicitly state the limitations of your work by adding “the limitations of this work are …. “

Please provide the full link or accession number to the data in the data availability statement

"In this study, we found that ..." or similar at line 77

Author summary Lines 118-121 should be toned down by adding “our results suggest” or similar

Introduction Line 133 contains a typo and should be “specified”

Line 151 please rephrase “have their health followed”

Line 247 please change “hypertensive” to “with hypertension”

Funding information can be removed from the main text and moved to the financial statement in the article meta-data

References- please remove all iterations of “Internet”

STROBE checklist, please use section and paragraph numbers, rather than page numbers.

If no prespecified analysis plan exists, please make sure that the Methods section transparently describes when analyses were planned, and when/why any data-driven changes to analyses took place.

In the limitations part of the discussion, please add a sentence or two about the limited generalisability of the UK Biobank, largely due to the low representation from ethnic minorities in the study sample.

Comments from Reviewers:

Reviewer #2: Many thanks authors for their effort to improve the manuscript. I am satisfied with the response and the revision. No further issues needing attention.

Reviewer #3: The authors adequately addressed my primary concerns and questions. The additional data of 3 more years of follow-up greatly strengthen this paper further. I commend the authors for taking the steps to update the analyses to include this additional data.

[LINK]

---

## [Editor Report · Decision Letter 3]

26 Nov 2020

Dear Dr Dwyer, 

On behalf of my colleagues and the academic editor, Dr. Amanda Paluch, I am delighted to inform you that your manuscript entitled "Accelerometer measured physical activity and the incidence of cardiovascular disease: Evidence from the UK Biobank cohort study" (PMEDICINE-D-20-01924R3) has been accepted for publication in PLOS Medicine. 

PRODUCTION PROCESS

Before publication you will see the copyedited word document (within 5 business days) and a PDF proof shortly after that. The copyeditor will be in touch shortly before sending you the copyedited Word document. We will make some revisions at copyediting stage to conform to our general style, and for clarification. When you receive this version you should check and revise it very carefully, including figures, tables, references, and supporting information, because corrections at the next stage (proofs) will be strictly limited to (1) errors in author names or affiliations, (2) errors of scientific fact that would cause misunderstandings to readers, and (3) printer's (introduced) errors. Please return the copyedited file within 2 business days in order to ensure timely delivery of the PDF proof. 

If you are likely to be away when either this document or the proof is sent, please ensure we have contact information of a second person, as we will need you to respond quickly at each point. Given the disruptions resulting from the ongoing COVID-19 pandemic, there may be delays in the production process. We apologise in advance for any inconvenience caused and will do our best to minimize impact as far as possible.

EARLY VERSION

PRESS

PROFILE INFORMATION

Thank you again for submitting the manuscript to PLOS Medicine. We look forward to publishing it. 

Best wishes, 

Adya Misra, PhD

Senior Editor 

PLOS Medicine

plosmedicine.org